**Data Availability Statement:** All relevant data are within the paper.

**Funding:** This work was supported by the Korea Medical Device Development Fund grant funded by the Korea government (the Ministry of Science and

# Impaired pulmonary ventilation beyond pneumonia in COVID-19: A preliminary observation

**Shohei Inui** [1,2☯], **Soon Ho Yoon** [3,4☯], **Ozkan Doganay** [5], **Fergus V. Gleeson** [6,7], **Minsuok Kim** [8] *

**1** Departments of Radiology, Graduate School of Medicine, The University of Tokyo, Tokyo, Japan, **2** Departments of Radiology, Japan Self- Defense Forces Central Hospital, Tokyo, Japan, **3** Department of Radiology, UMass Memorial Medical Center, Worcester, MA, United States of America, **4** Department of Radiology, Seoul National University Hospital, Seoul National University College of Medicine, Seoul, Korea, **5** Healthy Science Institute, Ege University, Izmir, Turkey, **6** Department of Oncology, University of Oxford, Oxford, United Kingdom, **7** Department of Radiology, The Churchill Hospital, Oxford University Hospitals NHS Trust, Headington, United Kingdom, **8** Wolfson School of Mechanical, Electrical and Manufacturing Engineering, Loughborough University, Loughborough, United Kingdom

☯ These authors contributed equally to this work.
* m.kim@lboro.ac.uk

## Abstract

### Background

Coronavirus disease 2019 (COVID-19) may severely impair pulmonary function and cause hypoxia. However, the association of COVID-19 pneumonia on CT with impaired ventilation remains unexplained. This pilot study aims to demonstrate the relationship between the radiological findings on COVID-19 CT images and ventilation abnormalities simulated in a computational model linked to the patients' symptoms.

### Methods

Twenty-five patients with COVID-19 and four test-negative healthy controls who underwent a baseline non-enhanced CT scan: 7 dyspneic patients, 9 symptomatic patients without dyspnea, and 9 asymptomatic patients were included. A 2D U-Net-based CT segmentation software was used to quantify radiological futures of COVID-19 pneumonia. The CT image-based full-scale airway network (FAN) flow model was employed to assess regional lung ventilation. Functional and radiological features were compared across groups and correlated with the clinical symptoms. Heterogeneity in ventilation distribution and ventilation defects associated with the pneumonia and the patients' symptoms were assessed.

### Results

Median percentage ventilation defects were 0.2% for healthy controls, 0.7% for asymptomatic patients, 1.2% for symptomatic patients without dyspnea, and 11.3% for dyspneic patients. The median of percentage pneumonia was 13.2% for dyspneic patients and 0% for the other groups. Ventilation defects preferentially affected the posterior lung and worsened with increasing pneumonia linearly ($y = 0.91x + 0.99$, $R^2 = 0.73$) except for one of the nine

ICT, the Ministry of Trade Industry and Energy, the Ministry of Health & Welfare, Republic of Korea, the Ministry of Food and Drug Safety) (Project Number: 202011A03). The funders had no role in study design, data collection and analysis, decision to publish, or preparation of the manuscript.

**Competing interests:** I have read the journal's policy and the authors of this manuscript have the following competing interests: Soon Ho Yoon works as a chief medical officer in MEDICALIP Co. Ltd. All other authors do not have a conflict of interest to declare associated with this publication. This does not alter our adherence to PLOS ONE policies on sharing data and materials.

**Abbreviations:** ARDS, Acute respiratory distress syndrome; AUC, Area under the curve; COVID-19, Coronavirus disease 2019; CT, Computed tomography; CV, Coefficient of variation; FAN, Full-scale airway network; GGO, Ground-glass opacity; RT-PCR, Real-time reverse polymerase chain reaction; SARS-COV2, Severe acute respiratory syndrome coronavirus 2.

dyspneic patients who had disproportionally large ventilation defects (7.8% of the entire lung) despite mild pneumonia (1.2%). The symptomatic and dyspneic patients showed significantly right-skewed ventilation distributions (symptomatic without dyspnea: 0.86 ± 0.61, dyspnea 0.91 ± 0.79) compared to the patients without symptom (0.45 ± 0.35). The ventilation defect analysis with the FAN model provided a comparable diagnostic accuracy to the percentage pneumonia in identifying dyspneic patients (area under the receiver operating characteristic curve, 0.94 versus 0.96).

## Conclusions

COVID-19 pneumonia segmentations from CT scans are accompanied by impaired pulmonary ventilation preferentially in dyspneic patients. Ventilation analysis with CT image-based computational modelling shows it is able to assess functional impairment in COVID-19 and potentially identify one of the aetiologies of hypoxia in patients with COVID-19 pneumonia.

## Introduction

COVID-19 causes a varying degree of dyspnea and hypoxemia [1]. Severe hypoxemia in COVID-19 can progress into acute respiratory failure, and this is the main cause of mortality [2]. Classically, severe hypoxemia in acute respiratory distress syndrome (ARDS) is associated with a large amount of non-aerated lung tissue on computed tomography (CT) [3], but a subset of COVID-19 patients with respiratory failure have profound hypoxemia but a minimal degree of non-aerated lung tissue on CT [4]. The manifestation of hypoxia not explained by CT has received significant attention as a significant clue to the pathophysiology of disease in COVID-19.

One of the pathological processes of COVID-19 is vascular damage, with microvascular thrombosis and ventilation-perfusion mismatch in the non-injured lung resulting in hypoxia [5], with up to 30% of critically-ill patients with COVID-19 having pulmonary embolic disease on CT [6]. The ventilation-perfusion mismatch in COVID-19 occurring due to hypoxic pulmonary vasoconstriction [5] and widespread microvascular thrombosis [7] associated with prothrombotic tendency [8].

This current understanding was based on the dichotomization of COVID-19-afflicted lung into normal-looking lung (aerated lung) and abnormal-looking lung with pulmonary opacities (non-aerated lung) on CT. The former subset has been regarded as having normal ventilation, but it is currently unknown whether ventilation is normal or impaired in the aerated lung on CT. Functional pulmonary ventilation imaging would be a direct way to observe pulmonary ventilation but has been rarely attempted in COVID-19 because the procedures required for functional imaging, specifically because the resulting aerosolization increases the risk of infection of staff [9]. Recent studies based on conventional CT imaging without using inhaled gas have demonstrated that regional ventilation computed by mathematical modeling correlates with conventional pulmonary ventilation imaging [10, 11] and is applicable to COVID-19 [12]. This study aimed to investigate ventilation abnormalities on chest CT images in COVID-19.

## Materials and methods

This study was conducted with the approval of our institutional ethics review board (Japan Self-Defense Forces Central Hospital, Approval Number 02–053). Written informed consent

was waived because of the retrospective nature of the study. The privacy of all patients was protected.

## Study population

We used a previously reported study sample containing data that has been published previously, regarding CT image analysis by radiologists [13]. This study used CT images from the part of the study population for ventilation modeling. We applied the following inclusion criteria: (a) admitted to the Japan Self-Defense Forces Central Hospital (Tokyo, Japan) from 2020/01/30 to 2020/02/28, (b) proved either to have COVID-19 by at least one positive result of real-time reverse polymerase chain reaction (RT-PCR) or not to have COVID-19 infection by two negative results of RT-PCR, (c) a chest CT scan performed on the same day of the RT-PCR. We excluded patients who had a smoking history or any comorbid lung disease to avoid the potential influence of smoking or comorbid disease on pulmonary ventilation assessment. Patients whose chest CT image had severe motion artifacts were also excluded.

From the Diamond Princess cohort, we included three types of patients with COVID-19: (1) asymptomatic patients, (2) symptomatic patients without dyspnea, (3) symptomatic patients with dyspnea. We considered patients complaining of shortness of breath as those having dyspnea. A study coordinator randomly selected patients from each group. A total of 29 cases (median age, 59 years; 16 men and 13 women) were included in this study (Fig 1). Twenty-five of them, 16 symptomatic and 9 asymptomatic patients had COVID-19 infection, and 4 patients did not have COVID-19 infection. Among the 16 symptomatic patients, 9 patients complained dyspnea, and 7 patients did not have dyspnea but complained of other mild symptoms (e.g., fever, cough, headache, lethargy, or gastrointestinal symptoms).

The study coordinator reviewed the medical records to collect the clinical and imaging data. The following items were extracted from the medical records: height, weight, body mass index (BMI), smoking history, comorbidities, and symptoms when the RT-PCR was performed. All analysis was performed blinded to the patients' symptoms and RT-PCR status.

## Chest CT acquisition and image analysis

Unenhanced chest CT scans were performed using a six-row multidetector CT scanner (SOMATOM Emotion 6; Siemens, Erlangen, Germany) with the following parameters:

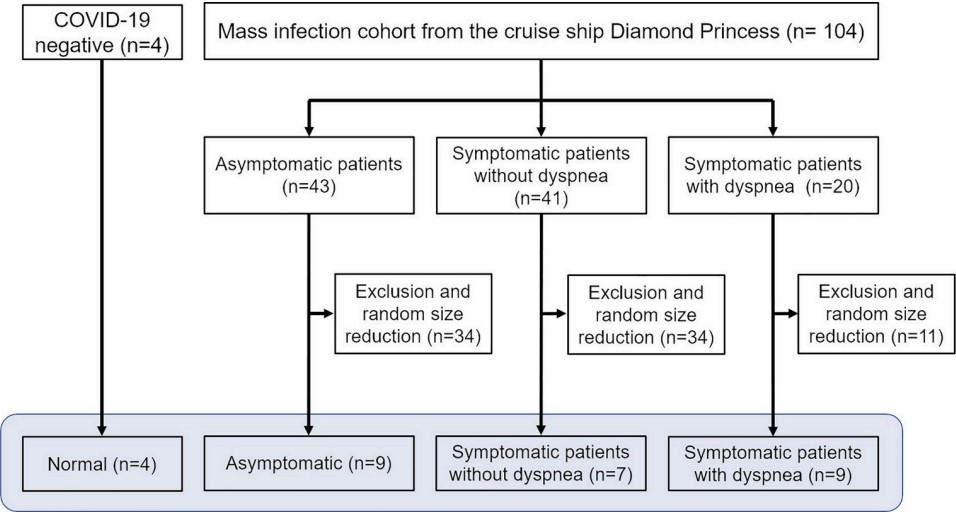

**Fig 1. Schematic diagram of the COVID-19 CT data collection for FAN flow modeling analysis.**

tube voltage, 130 kVp; effective current, 95mA; collimation, 6 x 2 mm, helical pitch, 1.4; field of view, 38 cm; matrix size, 512 x 512. A 1.0-mm thick slice with a 1.0-mm slice increment was reconstructed before analysis. Scanning was performed in the supine position at full inspiration. All patients underwent CT scan within 4 days after symptom onset for symptomatic patients, early to intermediate phase for radiologic manifestations of COVID-19 [14].

CT image analysis data was performed as per the previously published reports for the patients included in this study [13]. Evaluation included the following parameters [15]: the presence or absence of pure ground-glass opacity (GGO), crazy-paving pattern (GGO with inter- or intralobular septal thickening), consolidation, findings compatible with organizing pneumonia (i.e., consolidation with volume loss, subpleural curvilinear lines, and/or reversed halo sign) [16], and vascular enlargement inside the opacities. The number of affected lobes and anterior or posterior predominance was also recorded.

## Ventilation modeling

A CT image-based full-scale airway network (FAN) flow model was used to compute the dynamic ventilation. Patient-specific geometry of lobe surfaces, large airway structure, and parenchymal tissue density data was acquired from CT images. We used a branch-growing algorithm to generate small airways within the lobes [17, 18]. A 2D U-Net-based CT analysis software dedicated to COVID-19, MEDIP COVID19 (Medical IP, Seoul, Korea), was used to automatically segment the lesion masks of pneumonia from the CT images. The 2D U-Net was developed using 49,830 positive (pneumonia mask) and negative CT image slices from 176 CT scans of COVID-19 and provided the intraclass correlation coefficient and dice similarity score of 0.99 and 0.78, each, between the network output and reference mask in the dataset [19]. The lesion masks were mapped onto the FAN geometry to assess dynamic flow rate and gas concentration in the lungs. We assumed the degeneration of epithelial and smooth muscle cells in acini induced by COVID-19 inflammation caused significant loss of wall compliance. This consequently resulted in flow reduction in the associated airways. Tissue density assessed by using Hounsfield units from CT was used to determine the initial acinar volumes. To simulate the dynamic breathing in the FAN, we imposed a 0.2 Hz sinusoidal pleural pressure wave varying in a physiological range (maximum: -490 Pa, minimum: -890 Pa) on the acini surfaces [20].

Under an assumption of the insignificant inertial force during the normal breathing cycle, the flow in a single airway compartment can be calculated as

$$Q_d = \frac{P - P_d}{R} + \frac{C}{2}\left(\frac{dP}{dt} + \frac{dP_d}{dt}\right),$$ (1)

were $Q_d$ is the flow rate in an airway, $P$ and $P_d$ are the nodal pressures, and $R$ and $C$ are the airway resistance and compliance, respectively. If we assume acinar deformation over time $t$ is isotropic, the equation of acinar dynamics is formulated as

$$I\frac{d^2 V_a}{dt^2} + R_a\frac{dV_a}{dt} + \frac{V_a}{C_a} = P_a - P_{pl},$$ (2)

where $I$ is the inertance of acinar motion, $V_a$ is the volume of an acinus, $R_a$ is the resistance of acinar deformation, $C_a$ is the acinar compliance, $P_a$ and $P_{pl}$ are the intra-acinar pressure and the pleural pressure, respectively. The details of FAN modeling and validation of ventilation maps has been reported in previous studies [10, 11, 21, 22].

### Data analysis

Analysis of the CT scans related to visual description and ventilation modeling was then performed by grouping patients depending on their COVID-19 symptoms. We compared the relationships between the anatomical features (i.e., pneumonia) obtained using MEDIP COVID-19 and functional features (i.e., ventilation parameters) computed by the FAN model. We used two terms associated with the ventilation efficiency in the lungs: (i) ventilation heterogeneity was assessed with the coefficient of variation (CV), and (ii) ventilation defect zone was defined where the gas concentration is lower than 10% of the mean ventilation in the lung. Spearman's correlation and histogram skewness were assessed to quantify the strength and association between the pneumonia and the ventilation distribution. Additionally, the receiver operating characteristic curve analysis was performed to investigate the diagnostic ability of study parameters (percentage pneumonia, ventilation heterogeneity, and percentage ventilation defect) to classify symptomatic patients. We calculate the percentage pneumonia and the percentage ventilation defect based on their respective volume ratios to lung volume.

## Results

### Patient characteristics

Table 1 shows the demographic characteristics of all patients including total lung volume (cm$^3$), lung parenchymal CT attenuation (Hounsfield unit), percentage pneumonia, percentage ventilation defects, and ventilation CV. Age, sex distribution, height, weight and BMI. Participants with the RT-PCR test negative were slightly younger and lighter, and asymptomatic patients with COVID-19 were slightly older. Median total lung volumes (cm$^3$) of participants negative for COVID-19, asymptomatic patients, symptomatic patients without dyspnea, and dyspneic patients were 4651, 4757, 5493, and 4035 cm$^3$, respectively. Median lung parenchymal CT attenuations for those were -860, -856, -864, and -763 Hounsfield unit, respectively.

### CT image interpretation

Among the nine asymptomatic and CT positive patients, pure GGO was present in three patients, crazy-paving pattern in two, and consolidation in two. The pure GGO was observed in two, crazy-paving pattern in two, and consolidation in two out of seven mild symptomatic patients without dyspnea. The appearance of COVID-19 in CT was more obvious in symptomatic patients with dyspnea: the pure GGO, crazy-paving pattern and consolidation were found in four, five and seven out of nine patients, respectively. Median number of affected lobes in dyspneic patients were five and were zero in the asymptomatic patients and the symptomatic patients without dyspnea. In all but one of the cases that had CT abnormalities, the opacities showed a posterior predominance.

### Pneumonia and ventilation defects

Fig 2 demonstrates the pneumonia mask identified by MEDIP COVID19 and the FAN modelled ventilation defect in normal and patients with COVID-19. Median percentage pneumonia in dyspneic patients was 13.2% of the entire lung, whereas that in the other three groups was 0%, although upper quartiles were 0%, 1.3%, and 3.5%, respectively (Table 1). Median ventilation defects were 0.2% and 0.7% for participants without COVID-19 and asymptomatic patients with COVID-19 and increased to 1.2% and 11.3% for symptomatic patients without and with dyspnea, respectively. The percentage pneumonia, percentage ventilation defect, and ventilation CV (i.e., heterogeneity) were greater (i) in symptomatic patients than in asymptomatic patients (Fig 3A) and (ii) in dyspneic patients than in symptomatic patients without

**Table 1. Patient characteristics, anatomical, and functional CT findings.**

| | Negative for COVID-19 (n = 4) | Asymptomatic patients (n = 9) | Symptomatic patients without dyspnea (n = 7) | Dyspneic patients (n = 9) |
|---|---|---|---|---|
| **Median age (years)** | 53 (49–55) | 69 (57–71) | 60 (49–67) | 59 (45–68) |
| **Male:Female** | 2:2 | 4:5 | 4:3 | 6:3 |
| **Median height (m)** | 1.63 (1.62–1.65) | 1.62 (1.57–1.65) | 1.67 (1.63–1.74) | 1.65 (1.62–1.73) |
| **Median weight (kg)** | 58 (54–63) | 70 (62–75) | 70 (53–76) | 68 (63–72) |
| **Boddy mass index (BMI)** | 22.5 (17.7–24.2) | 26.0 (21.7–34.9) | 24.3 (19–26.7) | 25.1 (19.1–26.4) |
| **Respiratory rate (RR)** | 15 (14–18) | 16 (15–22) | 18 (14–24) | 21 (16–36) |
| **Saturation of percutaneous oxygen (%)** | 98 (98–99) | 97 (96–99) | 97 (93–99) | 94 (86–98) |
| **C-reactive protein (micro g/dL)** | 0.1* | 0.1 (0.1–0.4) | 1.3 (0.1–2.2) | 10.3 (2.1–16.7) |
| **Whole blood cells (per microliter)** | 6150* | 5630 (4990–6210) | 4881 (4503–5365) | 5822 (5290–8204) |
| **Lymphocyte (%)** | 20.7* | 27.4 (21.4–37.1) | 24.8 (21.9–27.4) | 12.1 (8.5–16.2) |
| **Lymphocyte (per microliter)** | 1273* | 1543 (1320–2252) | 1098 (996–1255) | 911 (705–943) |
| **Lactate dehydrogenase (IU/L)** | 158* | 182 (175–190) | 215 (183–252) | 305 (249–435) |
| **D-dimer (micro g/mL)** | 0.5* | Not measured | 0.6 (0.3–0.6) | 1.1 (0.8–1.8) |
| **Partial pressure of arterial O2 (mmHg)** | Not measured | Not measured | Not measured | 73.4 (68.9–79.6) |
| **Median lung volume (cm³)** | 4651 (4588 to 4735) | 4757 (4074 to 5101) | 5493 (5057 to 5547) | 4035 (3580 to 4538) |
| **Median average lung attenuation (HU)** | -860 (-866 to -849) | -856 (-861 to -851) | -864 (-881 to– 823) | -763 (-794 to -738) |
| **Percentage pneumonia in total lungs (non-aerated lung, %)** | 0.0 (0.0 to 0.0) | 0.0 (0.0 to 1.3) | 0.0 (0.0 to 3.5) | 13.2 (8.1 to 19.1) |
| **Aerated lung in total lungs (%)** | 100 (100 to 100) | 100 (99 to 100) | 100 (96.5 to 100) | 86.8 (80.9 to 91.9) |
| **Ventilation defects in total lungs (%)** | 0.5 (0.2 to 1.2) | 0.7 (0.3 to 2.6) | 1.2 (0.5 to 5.3) | 11.3 (7.8 to 24.0) |
| **Lung ventilation coefficient of variation** | 0.45 (0.44 to 0.50) | 0.43 (0.41 to 0.59) | 0.60 (0.54 to 0.66) | 0.63 (0.57 to 0.82) |
| **Ground-glass opacity (GGO)** | 0 | 3 (33%) | 2 (29%) | 4 (44%) |
| **Crazy-paving pattern** | 0 | 2 (22%) | 2 (29%) | 5 (56%) |
| **Consolidation** | 0 | 2 (22%) | 2 (29%) | 7 (78%) |
| **Number of lobes affected** | 0 | 0 (0–1) | 0 (0–5) | 5 (2–5) |

Data in parenthesis indicate interquartile range.

*One of the four patients underwent laboratory tests.

dyspnea (Fig 3B). When the lungs were divided into anterior and posterior halves, ventilation defects predominantly appeared in the posterior lungs similar to pneumonia distribution regardless of the presence of symptoms or dyspnea (Fig 3C and 3D).

Fig 4 illustrates the normalized ventilation distribution in the normal, asymptomatic, symptomatic and symptomatic with dyspnea groups. The symptomatic and dyspneic patients had the right-skewed ventilation distribution toward poor ventilation compared to the patients without symptom. The mean and standard deviation of skewness were $0.45 \pm 0.35$, $0.86 \pm 0.61$, $0.91 \pm 0.79$ for asymptomatic, symptomatic without and symptomatic with dyspnea groups, respectively. When ventilation defect against pneumonia was plotted among patients having percentage pneumonia of greater than 1%, the linear relationship was observed ($y = 0.91x + 0.99$, $R^2 = 0.73$; Fig 5). However, 1 of 9 dyspneic patients had disproportionally larger ventilation defects in 7.8% of the entire lung, whilst pneumonia was confined to 1.2% of the lung.

Fig 6A and 6B illustrate the analyses of pulmonary ventilation on a lobar level. The correlation between the percentage lobar non-pneumonia on CT and the percentage ventilation was significant ($\rho = 0.68$, $p < 0.001$). Similarly, the linear relationship between percentage lobar pneumonia and lobar ventilation CV was significant for a pneumonia ratio that was larger than 1% ($\rho_{LP} = 0.71$, $p < 0.001$). Fig 6C and 6D show the connection between sensitivity and specificity to assess the diagnostic ability of percentage pneumonia, ventilation CV and

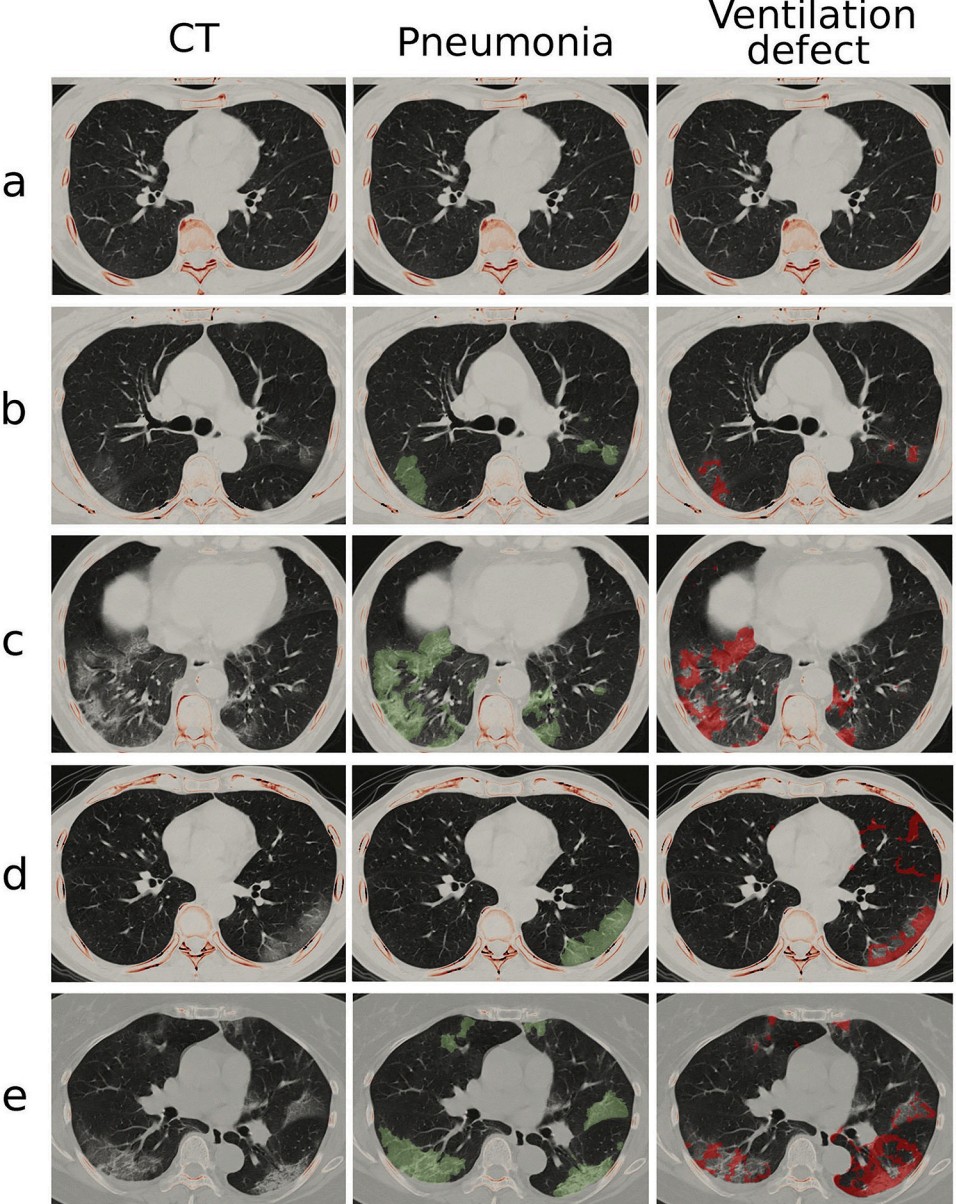

**Fig 2. Pneumonia mask and ventilation defect on CT images.** The cohort is categorized into (a) normal, (b) COVID-19 asymptomatic, (c) symptomatic without dyspnea and (d-e) symptomatic with dyspnea groups. The middle and right columns show pneumonia mask (green-colored layer) and FAN modeled ventilation defect (red-colored layer) on a transverse plane, respectively.

percentage ventilation defect as a classifier of symptoms and dyspnea. The area under the curve (AUC) for the presence of symptoms and dyspnea, the percentage ventilation defect (AUCs, 0.90 and 0.94, each) provided similar diagnostic performances with percentage pneumonia (AUCs, 0.84 and 0.96, each).

## Discussion

In the present study, we have modeled poor ventilation in COVID-19 lungs associated with severe acute respiratory syndrome coronavirus 2 (SARS-COV-2) infection. None of the

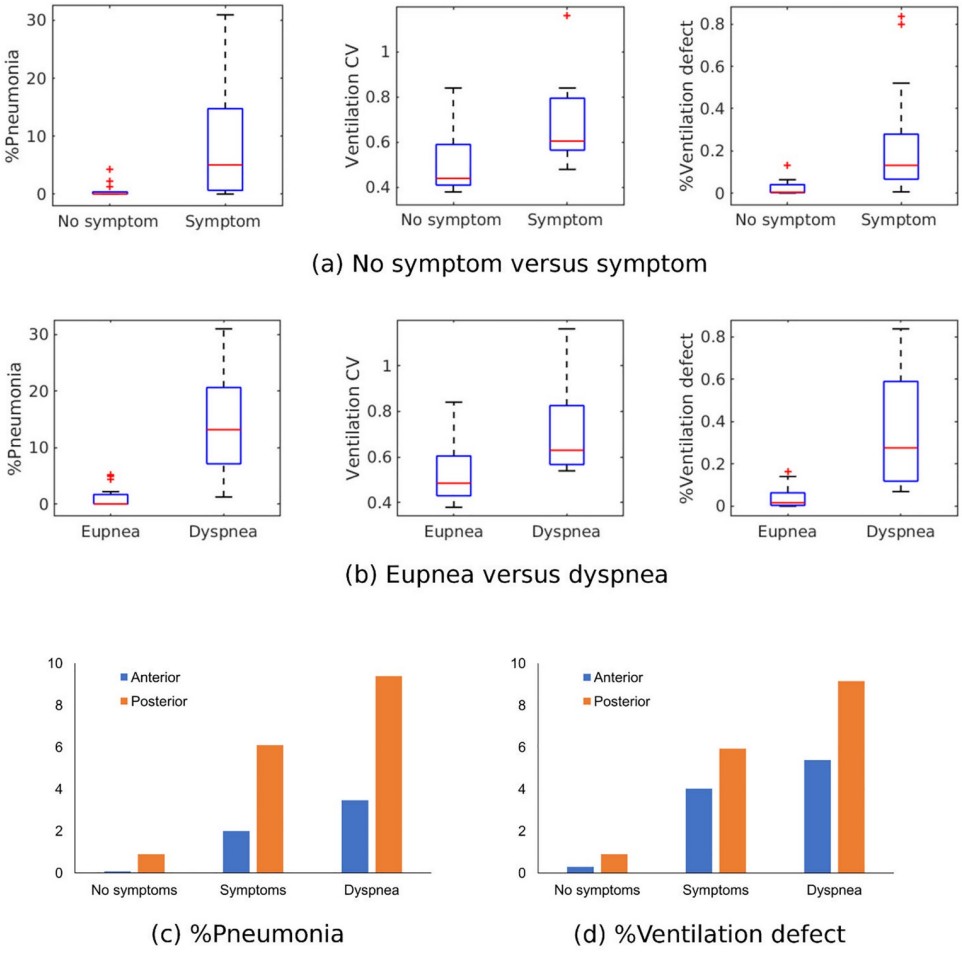

**Fig 3.** Comparison of %pneumonia, ventilation coefficient of variation (CV) and %ventilation defect between (a) no symptom and symptom groups; (b) eupnea and dyspnea groups. Comparison of (c) %pneumonia and (d) %ventilation defect between anterior and posterior lungs.

patients in this study had ever smoked, and comorbid pulmonary diseases were absent. Poorly-ventilated areas were minimally present in the negative control and asymptomatic patients in comparison to the symptomatic patients. The linear relationship between the lobar percentage non-pneumonia (normal-looking lung) ventilation ratio was similar to that in prior reports of gas ventilation imaging [10, 11]. The poorly-ventilated areas mapped the distribution of pneumonia and predominated in the posterior lung in keeping with reports of SARS-COV-2 involvement [17]. Reports of postmortem lung specimens infected by SARS-COV-2 show a varying degree of alveolar and bronchial inflammation [18]. Grossly-normal but poorly-ventilated CT areas result from a mild degree of alveolar inflammation or lower airway damage hampering pulmonary ventilation [20] beyond CT visual resolution preventing identification of pulmonary opacities.

We used the FAN model to determine the effect of COVID-19 pneumonia on peripheral airway flow restriction beyond the direct visual identification on CT. The dysfunction of small airways alters the dynamic flow characteristics in neighboring and proximal airways. Multiple studies have reported this network flow behavior in asymmetric airways [23, 24]. They have demonstrated that the local impairment of pulmonary function may provoke significant large

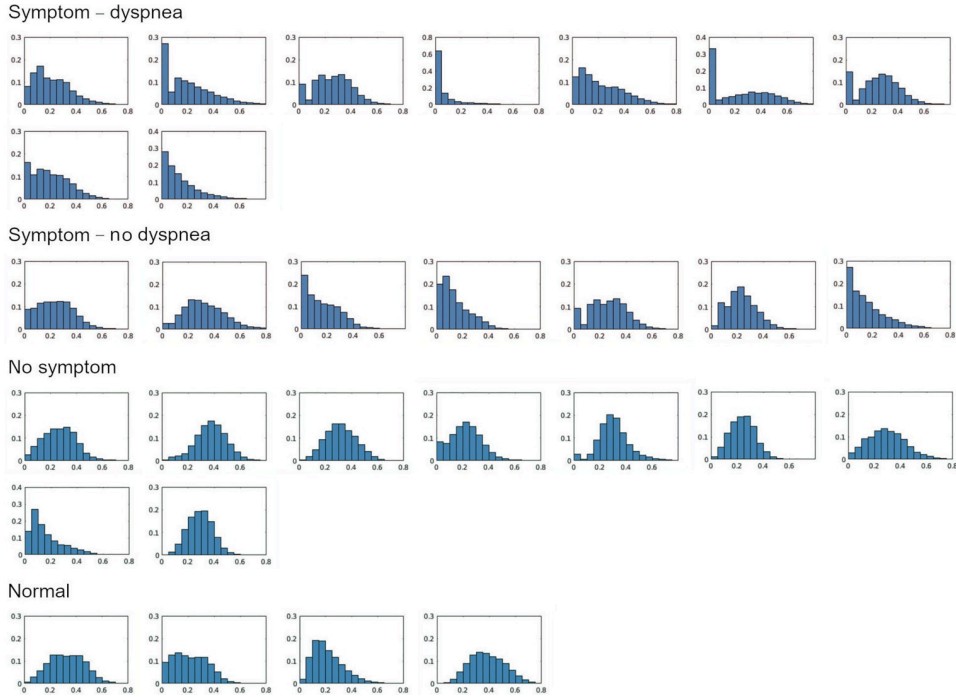

**Fig 4. Lung ventilation distribution in normal, asymptomatic, symptomatic without dyspnea and with dyspnea cases.**

scale ventilation defects. In this study, the FAN model included the simulation of acinar flow alteration due to structural destruction in areas of SARS-COV2 infection. The local airway flow changes affected the flows in other parts of the airways within the network system. The integrated acinar dysfunction and resultant ventilation defects computed in the FAN model were in good agreement with the percentage pneumonia. Also, we found weakened/delayed ventilation, which might cause increased ventilation heterogeneity even though there was minimal visualized pneumonia.

Specifically reviewing our results and dyspnea, we are unable to comment on whether vascular pathologies co-existed in the poorly-ventilated areas, particularly for normal-looking areas on CT images. The poor ventilation without gross lung parenchymal abnormalities on CT images may result from decreased perfusion due to pulmonary embolism, microvascular thrombosis, or pulmonary hypoxic vasoconstriction. Potentially the severe hypoxemia in patients with minimal or non-severe COVID-19 pneumonia on CT not explained by vascular pathology [5], may be due to the poorly-ventilated areas having preserved or increased perfusion due to vascular dysregulation resulting in intrapulmonary shunting and ventilation-perfusion mismatch. Recent studies showed more than a third of patients discharged from a hospital for COVID-19 were symptomatic after three months. Most commonly experiencing dyspnea and fatigue possibly due to impaired ventilation efficiency [25, 26].

CT pulmonary angiography with FAN ventilation modeling analysis in COVID-19 may reveal the origin of hypoxemia and help patient-specific management. The prior published case report of FAN ventilation modeling in a dyspneic patient with COVID-19 with pulmonary embolism on a CT pulmonary angiogram excluded as the cause, had a preferential anterior ventilation defect in 20.4% of the entire lung but with a percentage pneumonia involvement of only 0.9% is supportive of this suggestion [12]. Additionally pulmonary microvascular thrombosis resulting in perfusion defects assessed using dual energy CT has also

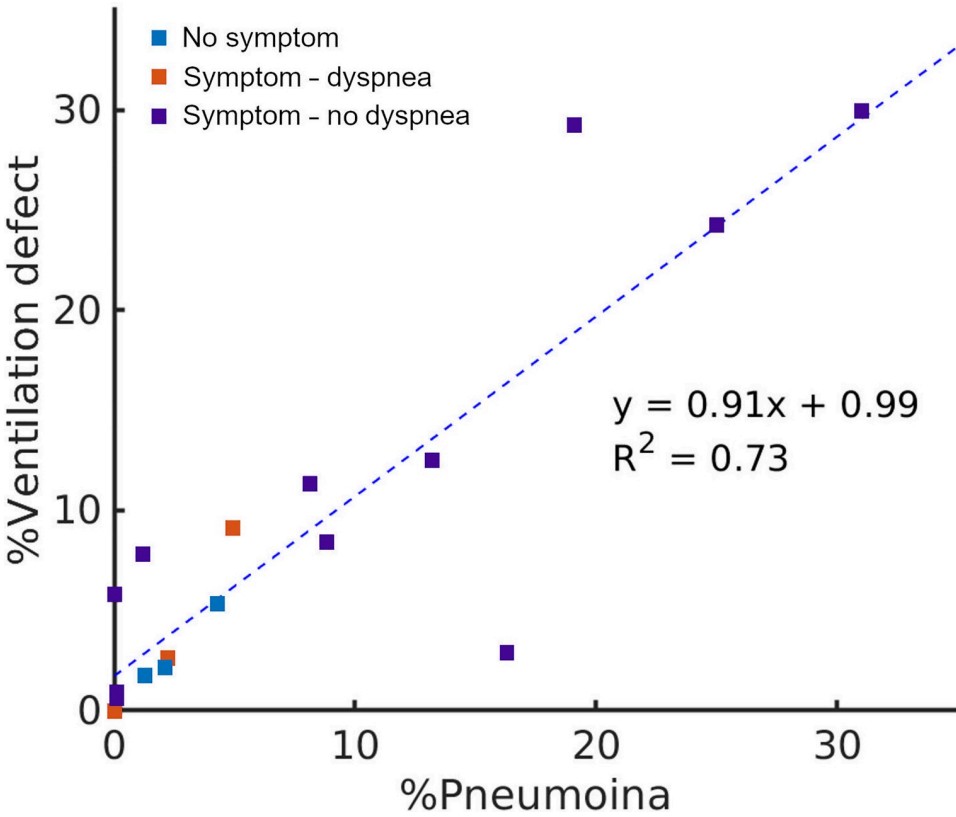

**Fig 5. Relationship between %pneumonia (%; x-axis) and %ventilation defect (%; y-axis).**

previously been reported to be useful in patients with COVID-19 [27] and this combined with FAN ventilation modeling may help assess whether vascular pathology or intrapulmonary shunting or a combination of both may be the cause of hypoxemia, and whether aerated or non-aerated lungs with normal or impaired ventilation dominate when considering positive pressure ventilation and whether supine or prone positioning are appropriate.

This retrospective pilot study has several limitations. We have shown that normal-looking aerated lungs on non-contrast CT scans in patients with COVID-19 pneumonia have ventilation defects using FAN modelling. These defects were most prevalent in patients experiencing dyspnea. The ventilation defects typically increased as the percentage of pneumonia increased, but some of the patients had disproportionally large ventilation defect areas with minimal pneumonic change on CT. We have demonstrated a case with a relatively large ventilation defect area compared to the degree of pneumonia identified on CT. This might indicate a cause of breathlessness in patients with less severe pneumonia on CT. However, the number of cases presented in this pilot study is small. Additional studies involving larger cohorts are necessary to confirm this. Although prior publications of FAN modeling have demonstrated its accuracy, reporting significant correlation with pulmonary function imaging in patients with COPD [10, 11], our ventilation modeling results have not been validated by gas ventilation imaging, and there was no absolute cutoff universally applicable to define ventilation defects. Also, as the patients included only underwent a non-contrast CT scan, pulmonary embolic disease as a potential cause of their symptoms could not be assessed. Additionally, the current FAN ventilation modeling requires considerable time and resources to calculate, so this approach could not currently be introduced into clinical practice.

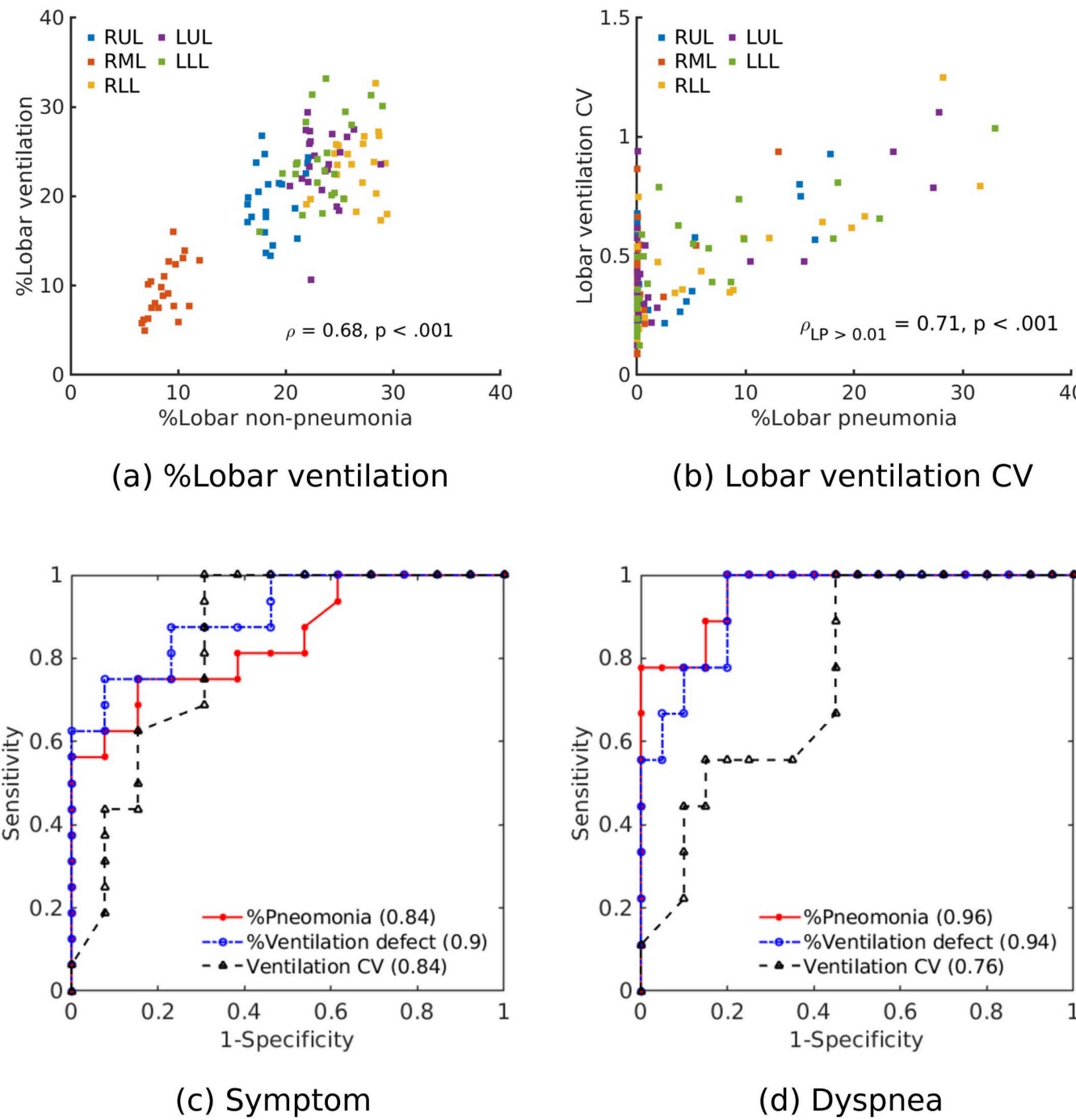

**Fig 6.** Relationship between (a) %lobar ventilation and %lobar non-pneumonia; (b) ventilation CV to %lobar non-pneumonia. Receiver operating characteristic (ROC) curves to classify (c) symptoms and (d) dyspnea. The area under the curves for %pneumonia, %ventilation defect and ventilation CV are shown in the legend.

## Conclusions

In conclusion, this study has used a CT image-based FAN model to investigate impaired ventilation induced by pneumonia in COVID-19 lungs. The FAN model was potentially capable of deriving regional ventilation and dynamic airflow impairments in dyspnea and symptomatic

patient groups. Although the FAN model has focused on the computational evaluation of ventilation defects, its sensitivity and specificity were comparable to the extent of pneumonia identified on CT. In addition to using ventilation abnormalities identified with the FAN model, including perfusion from CTPA scans and the consequent ability to model gas-exchange could potentially help with understanding the pathophysiology and profound hypoxia that symptomatic patients experience.

## Author Contributions

**Conceptualization:** Soon Ho Yoon, Minsuok Kim.

**Data curation:** Shohei Inui, Soon Ho Yoon, Minsuok Kim.

**Formal analysis:** Shohei Inui, Soon Ho Yoon, Minsuok Kim.

**Investigation:** Shohei Inui, Soon Ho Yoon, Minsuok Kim.

**Methodology:** Shohei Inui, Soon Ho Yoon, Minsuok Kim.

**Project administration:** Soon Ho Yoon, Minsuok Kim.

**Resources:** Minsuok Kim.

**Software:** Soon Ho Yoon, Minsuok Kim.

**Supervision:** Soon Ho Yoon, Fergus V. Gleeson, Minsuok Kim.

**Validation:** Shohei Inui, Soon Ho Yoon, Ozkan Doganay, Fergus V. Gleeson, Minsuok Kim.

**Visualization:** Minsuok Kim.

**Writing – original draft:** Shohei Inui, Soon Ho Yoon, Minsuok Kim.

**Writing – review & editing:** Shohei Inui, Soon Ho Yoon, Ozkan Doganay, Fergus V. Gleeson, Minsuok Kim.

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
