## [Decision Letter · Decision Letter 0]

2 Dec 2021

PONE-D-21-28780Impaired Pulmonary Ventilation Beyond Pneumonia in COVID-19: A preliminary observationPLOS ONE

Dear Dr. Kim,

Thank you for submitting your manuscript to PLOS ONE. After careful consideration, we feel that it has merit but does not fully meet PLOS ONE’s publication criteria as it currently stands. Therefore, we invite you to submit a revised version of the manuscript that addresses the points raised during the review process. As can be noted from the enclosed reviews, comments were generally positive about this manuscript, yet both reviewers had several concerns. Specifically, both reviewers commented on the impact of outliers within the study and the need to carefully address this issue. In addition, data presentation can be improved as suggested.  

We look forward to receiving your revised manuscript.

Kind regards,

Ruud AW Veldhuizen

Academic Editor

PLOS ONE

Journal Requirements:

“I have read the journal's policy and the authors of this manuscript have the following competing interests:

Soon Ho Yoon works as a chief medical officer in MEDICALIP Co. Ltd. All other authors do not have a conflict of interest to declare associated with this publication”

Reviewers' comments:

Reviewer's Responses to Questions

**Comments to the Author**

1. Is the manuscript technically sound, and do the data support the conclusions?

Reviewer #1: Yes

Reviewer #2: Partly

2. Has the statistical analysis been performed appropriately and rigorously? 

Reviewer #1: Yes

Reviewer #2: Yes

3. Have the authors made all data underlying the findings in their manuscript fully available?

Reviewer #1: Yes

Reviewer #2: No

4. Is the manuscript presented in an intelligible fashion and written in standard English?

Reviewer #1: Yes

Reviewer #2: Yes

5. Review Comments to the Author

Reviewer #1: The authors presented a retrospective study of analyzing the relationship of ventilation defect with pneumonia by CT-based FAN flow model for COVID-19 patients. Their results demonstrated the availability to assess functional impairment in COVID-19 from the integration of clinical imaging with computational modelling. In addition, the diagnostic accuracy in identifying dyspneic patients by the ventilation defect analysis with the FAN was also satisfying. The study addresses an important issue and the results are inspiring for clinicians. Some questions still remain to be cleared.

1, Since the FAN flow model is the foundational methodology of this work, the authors should provide more details about the flow calculation (may be in the supplementary material), such as the flow equations, model hypothesis, simulation method and so on. I understand that the modeling details have been described in their published studies as they referred to. However, it is better to explain the basic methodology in this paper for the readers who know the FAN model for the first time.

2, Is there a better way to display the results in Fig. 4? For example, the panels can be arranged based on the different groups.

3, The authors declared that there was an outlier (x=1.2%, y=7.8%) in Fig. 5. However, there are two more outliers in the figure, one is around (18%, 3%), the other is around (19%, 29%). Is there any analysis about this result?

4, In Fig. 6a & b, the authors showed the percentage ventilation and ventilation CV with different lobes indicated by different colors. The correlations showed in the text, however, were given by the total data. How are the correlations between ventilation with pneumonia on different lobes?

5, Some typos. For example, Page 16, Line 355, 356, extra space.

Reviewer #2: The authors reported interesting findings from a pilot study performed on patients with COVID-19 (n=25) and healthy controls (n=4). They provided a post-processing analysis to quantify regional lung ventilation. Dyspnoic CVOID-19 patients had both higher anatomical involvement and larger ventilation defect as compared to non-dyspnoic patients. A direct correlation linked anatomical involvment with ventilation defect; a single patient had a moderate non-concordance (eg higher ventilation defect as expected by the anatomical involvmente). This is a well performed stimulating study, very appealing from a technical point of view. My main concern regards the authors' hypothesis, that though intriguing, should be partly downgraded in light of the study result; indeed, the evident finding from the present study is the robustness of the FAN technique in the evaluation of COVID-19 pneumonia as compared to the traditional anatomical evalutaion. However, the possibility of FAN to provide additional information, though plausible, relys on a single outlier and should be more carefully reported.

Minor comments:

- If available, some few more data could be reported (i.e. dimer level, CRP, P/F etc. and their correlation with CT findings)

- Figure 5 could be displayed with dots of different colour according to the underlying study group (asymtpomatic, dyspnoic etc)

6. PLOS authors have the option to publish the peer review history of their article (what does this mean?). If published, this will include your full peer review and any attached files.

Reviewer #1: No

Reviewer #2: No

---

## [Author Response · Author response to Decision Letter 0]

29 Dec 2021

Journal Requirements:

>> The revised manuscript meets PLOS ONE’s style requirements.

>> We are able to open-source the main code, which will allow researchers to see the structure of how it runs and links with the other parts of the code. (Ref. 21, Github link and software version). We have also put the functions and libraries for model boundaries, IOs and data analysis in a public repository. This should allow researchers to start to replicate the work. Because there is commercial interest and involvement of more than one large organization in the development of the code, the intellectual property ownership is hard to disambiguate, and without extensive permissions the only thing we can’t open-source immediately are some of the functions. To get around this we have already started a project to work on an open-source version of this software, and plan to release in about 1-2 years.

>> We have stated followings for the financial disclosure.

------------

This work was supported by the Korea Medical Device Development Fund grant funded by the Korea government (the Ministry of Science and ICT, the Ministry of Trade Industry and Energy, the Ministry of Health & Welfare, Republic of Korea, the Ministry of Food and Drug Safety) (Project Number: 202011A03). The funders had no role in study design, data collection and analysis, decision to publish, or preparation of the manuscript.

“I have read the journal's policy and the authors of this manuscript have the following competing interests:

Soon Ho Yoon works as a chief medical officer in MEDICALIP Co. Ltd. All other authors do not have a conflict of interest to declare associated with this publication”

>> We have included following Competing Interests statement in the cover letter.

------------

I have read the journal's policy and the authors of this manuscript have the following competing interests: Soon Ho Yoon works as a chief medical officer in MEDICALIP Co. Ltd. All other authors do not have a conflict of interest to declare associated with this publication. This does not alter our adherence to PLOS ONE policies on sharing data and materials.

>> All relevant data are within the manuscript. The institutional IRB allowed the analysis of CT images for this study solely, so CT images used in this study cannot be shared based on the IRB approval. 

>> All relevant data are within the manuscript. The institutional IRB allowed the analysis of CT images for this study solely, so CT images used in this study cannot be shared based on the IRB approval.

Reviewers' comments:

Reviewer's Responses to Questions

Comments to the Author

Reviewer #1: The authors presented a retrospective study of analyzing the relationship of ventilation defect with pneumonia by CT-based FAN flow model for COVID-19 patients. Their results demonstrated the availability to assess functional impairment in COVID-19 from the integration of clinical imaging with computational modelling. In addition, the diagnostic accuracy in identifying dyspneic patients by the ventilation defect analysis with the FAN was also satisfying. The study addresses an important issue and the results are inspiring for clinicians. Some questions still remain to be cleared.

1, Since the FAN flow model is the foundational methodology of this work, the authors should provide more details about the flow calculation (may be in the supplementary material), such as the flow equations, model hypothesis, simulation method and so on. I understand that the modeling details have been described in their published studies as they referred to. However, it is better to explain the basic methodology in this paper for the readers who know the FAN model for the first time.

>> We have added governing equations of the FAN model in the manuscript.

------------

(Line 194-203)

Under an assumption of the insignificant inertial force during the normal breathing cycle, the flow in a single airway compartment can be calculated as

Q_d=(P-P_d)/R+C/2 (dP/dt+(dP_d)/dt), (1)

were Q_d is the flow rate in an airway, P and P_d are the nodal pressures, and R and C are the airway resistance and compliance, respectively. If we assume acinar deformation over time t is isotropic, the equation of acinar dynamics is formulated as

I (d^2 V_a)/(dt^2 )+R_a (dV_a)/dt+V_a/C_a =P_a-P_pl, (2)

where I is the inertance of acinar motion, V_a is the volume of an acinus, R_a is the resistance of acinar deformation, C_a is the acinar compliance, P_a and P_pl are the intra-acinar pressure and the pleural pressure, respectively.

2, Is there a better way to display the results in Fig. 4? For example, the panels can be arranged based on the different groups.

>> Thanks to the reviewer for this comment. We have updated Fig 4 with the newly arranged panels based on the different groups.

3, The authors declared that there was an outlier (x=1.2%, y=7.8%) in Fig. 5. However, there are two more outliers in the figure, one is around (18%, 3%), the other is around (19%, 29%). Is there any analysis about this result?

>> The current analysis includes all data points including outliers (without the outliers, y=0.89x + 2.06 and R2=0.94). The outliers demonstrate the existence of disagreement between structural abnormality (ex: % pneumonia) and functional impairment (ex: %ventilation) as shown in previous lung studies (Cerveri et al., Chest 2004; Rabe et al., Am J Respir Crit Care Med 2017; Yahaba et al., Eur J Radiol 2014). In this figure, we call attention to one outlier which shows large ventilation defects (functional impairment) even though minor pneumonia (structural abnormality). But as shown in the discussion, we have left its confirmation for follow-up studies with a larger cohort. 

4, In Fig. 6a & b, the authors showed the percentage ventilation and ventilation CV with different lobes indicated by different colors. The correlations showed in the text, however, were given by the total data. How are the correlations between ventilation with pneumonia on different lobes?

>> In this pilot study with a small number of samples, we used lobar parameters and present the correlation using all data points instead of comparing the lobar correlations. Similar lobar analyses have been presented in multiple previous lung image studies (Matin et al, Radiology 2016; Doganay et al., Eur Radiol 2019).

5, Some typos. For example, Page 16, Line 355, 356, extra space.

>> Thanks to the reviewer to find this. We have corrected those typos.

Reviewer #2: The authors reported interesting findings from a pilot study performed on patients with COVID-19 (n=25) and healthy controls (n=4). They provided a post-processing analysis to quantify regional lung ventilation. Dyspnoic CVOID-19 patients had both higher anatomical involvement and larger ventilation defect as compared to non-dyspnoic patients. A direct correlation linked anatomical involvment with ventilation defect; a single patient had a moderate non-concordance (eg higher ventilation defect as expected by the anatomical involvmente). This is a well performed stimulating study, very appealing from a technical point of view. My main concern regards the authors' hypothesis, that though intriguing, should be partly downgraded in light of the study result; indeed, the evident finding from the present study is the robustness of the FAN technique in the evaluation of COVID-19 pneumonia as compared to the traditional anatomical evalutaion. However, the possibility of FAN to provide additional information, though plausible, relys on a single outlier and should be more carefully reported.

>> We agree with the reviewer's comment. We have tried not to draw to strong conclusion from this small pilot study, and as a consequence of the reviewer’s suggestion, the followings have been added to the discussion.

------------

(Line 368-372) 

We have demonstrated a case with a relatively large ventilation defect area compared to the degree of pneumonia identified on CT. This might indicate a cause of breathlessness in patients with less severe pneumonia on CT. However, the number of cases presented in this pilot study is small. Additional studies involving larger cohorts are necessary to confirm this.

>> Additionally, conclusion section has now been reworded to address reviewer’s comments.

------------

(Line 383-391)

In conclusion, this study has used a CT image-based FAN model to investigate impaired ventilation induced by pneumonia in COVID-19 lungs. The FAN model was potentially capable of deriving regional ventilation and dynamic airflow impairments in dyspnea and symptomatic patient groups. Although the FAN model has focused on the computational evaluation of ventilation defects, its sensitivity and specificity were comparable to the extent of pneumonia identified on CT. In addition to using ventilation abnormalities identified with the FAN model, including perfusion from CTPA scans and the consequent ability to model gas-exchange could potentially help with understanding the pathophysiology and profound hypoxia that symptomatic patients experience. 

Minor comments:

- If available, some few more data could be reported (i.e. dimer level, CRP, P/F etc. and their correlation with CT findings)

>> We have added CRP Whole blood cells, Lymphocyte, Lactate dehydrogenase, D-dimer and Partial pressure of arterial O2 in the table 1.

- Figure 5 could be displayed with dots of different colour according to the underlying study group (asymtpomatic, dyspnoic etc)

>> Figure 5 has been updated with different colour according to the study group.

---

## [Decision Letter · Decision Letter 1]

13 Jan 2022

Impaired Pulmonary Ventilation Beyond Pneumonia in COVID-19: A preliminary observation

PONE-D-21-28780R1

Dear Dr. Kim,

We’re pleased to inform you that your manuscript has been judged scientifically suitable for publication and will be formally accepted for publication once it meets all outstanding technical requirements.

Kind regards,

Ruud AW Veldhuizen

Academic Editor

PLOS ONE

Additional Editor Comments (optional):

Reviewers' comments:

Reviewer's Responses to Questions

**Comments to the Author**

1. If the authors have adequately addressed your comments raised in a previous round of review and you feel that this manuscript is now acceptable for publication, you may indicate that here to bypass the “Comments to the Author” section, enter your conflict of interest statement in the “Confidential to Editor” section, and submit your "Accept" recommendation.

Reviewer #1: All comments have been addressed

Reviewer #2: All comments have been addressed

2. Is the manuscript technically sound, and do the data support the conclusions?

Reviewer #1: Yes

Reviewer #2: Yes

3. Has the statistical analysis been performed appropriately and rigorously? 

Reviewer #1: Yes

Reviewer #2: Yes

4. Have the authors made all data underlying the findings in their manuscript fully available?

Reviewer #1: Yes

Reviewer #2: Yes

5. Is the manuscript presented in an intelligible fashion and written in standard English?

Reviewer #1: Yes

Reviewer #2: Yes

6. Review Comments to the Author

Reviewer #1: (No Response)

Reviewer #2: Thank you for addressing in a proper way the Reviewers' comments. I have no further comments to be made to the articlr.

7. PLOS authors have the option to publish the peer review history of their article (what does this mean?). If published, this will include your full peer review and any attached files.

Reviewer #1: No

Reviewer #2: **Yes: **Luca Arcari

---

## [Editor Report · Acceptance letter]

17 Jan 2022

PONE-D-21-28780R1 

Impaired Pulmonary Ventilation Beyond Pneumonia in COVID-19: A preliminary observation 

Dear Dr. Kim:

I'm pleased to inform you that your manuscript has been deemed suitable for publication in PLOS ONE. Congratulations! Your manuscript is now with our production department. 

Kind regards, 

on behalf of

Dr. Ruud AW Veldhuizen 

Academic Editor

PLOS ONE